# Few-shot Learning for Feature Selection with Hilbert-Schmidt Independence Criterion

**Atsutoshi Kumagai**
NTT Computer and Data Science Laboratories
atsutoshi.kumagai.ht@hco.ntt.co.jp

**Tomoharu Iwata**
NTT Communication Science Laboratories
tomoharu.iwata.gy@hco.ntt.co.jp

**Yasutoshi Ida**
NTT Computer and Data Science Laboratories
yasutoshi.ida@ieee.org

**Yasuhiro Fujiwara**
NTT Communication Science Laboratories
yasuhiro.fujiwara.kh@hco.ntt.co.jp

## Abstract

We propose a few-shot learning method for feature selection that can select relevant features given a small number of labeled instances. Existing methods require many labeled instances for accurate feature selection. However, sufficient instances are often unavailable. We use labeled instances in multiple related tasks to alleviate the lack of labeled instances in a target task. To measure the dependency between each feature and label, we use the Hilbert-Schmidt Independence Criterion, which is a kernel-based independence measure. By modeling the kernel functions with neural networks that take a few labeled instances in a task as input, we can encode the task-specific information to the kernels such that the kernels are appropriate for the task. Feature selection with such kernels is performed by using iterative optimization methods, in which each update step is obtained as a closed-form. This formulation enables us to directly and efficiently minimize the expected test error on features selected by a small number of labeled instances. We experimentally demonstrate that the proposed method outperforms existing feature selection methods.

## 1 Introduction

Feature selection is a fundamental problem in machine learning that aims to find the subset of relevant features [7]. By extracting a small subset of features, we can interpret the characteristics of datasets, accelerate the learning processes of subsequent problems such as regression and classification, and eliminate the cost of collecting irrelevant or redundant features. Thanks to these beneficial properties, feature selection methods have been used in various applications such as biomarker discovery [52], document categorization [10], disease diagnosis [1], and drug development [32].

Many feature selection methods have been proposed. An unsupervised approach aims to find relevant features without label information [3, 30, 5]. Although it is useful, it has difficulty finding features that are responsible for predicting responses without labels. On the other hand, a supervised approach can find such features by using labeled instances [48, 34, 53, 54, 29]. Therefore, this paper focuses on supervised feature selection. Existing supervised feature selection methods often require many labeled instances to accurately select relevant features [21]. However, sufficient instances can be difficult to collect in many applications. For example, although feature selection is used for analysis in healthcare [36, 31], sufficient instances (e.g., patients) for prognosis analysis might be difficult to collect in a hospital. In marketing, when data scientists want to analyze sales by using past data in each area, sufficient instances might be difficult to collect from underdeveloped or new areas.

36th Conference on Neural Information Processing Systems (NeurIPS 2022).

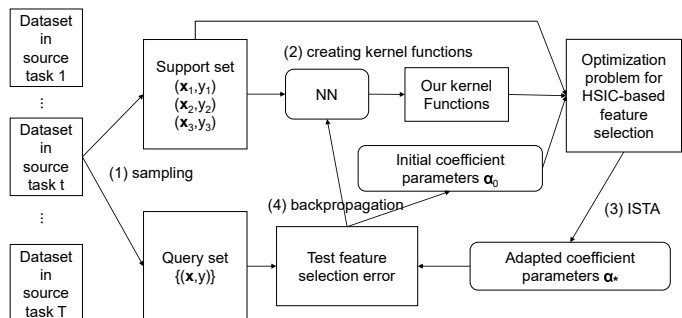

Figure 1: Our meta-learning procedure: (1) For each training iteration, we randomly sample a few labeled instances (support set) and test instances (query set) from a randomly selected source task. We assume that all tasks have the same feature space. (2) Kernel functions are inferred by the neural network that takes the support set as input, and the optimization problem for HSIC-based feature selection with the support set is defined. (3) We solve this problem by using ISTA, in which its initial coefficient parameters are treated as trainable parameters. (4) Test feature selection error in terms of HSIC is calculated with both the query set and adapted coefficient parameters, and it can be backpropagated to update both the neural network and initial coefficient parameters.

Even if many labeled instances are difficult to collect in a task of interest, called a target task, they might be obtainable in different but related tasks, called source tasks. For example, sufficient labeled instances might be obtainable from other hospitals or urban areas. When target and source tasks are related, we can transfer useful knowledge in source tasks to target ones [17].

In this paper, we propose a few-shot learning method for supervised feature selection. Few-shot learning is usually formulated as meta-learning, which learns how to learn from a few data using data in multiple tasks. Specifically, the proposed method meta-learns with labeled instances in multiple source tasks and uses the learned knowledge for feature selection with a small number of labeled instances in unseen target tasks. Figure 1 illustrates the overview of our meta-learning procedure.

In feature selection, the dependency between each feature and response needs to be measured. To this end, we use the Hilbert-Schmidt Independence Criterion (HSIC), which is a non-parametric kernel-based independence measure [15]. Since HSIC can measure the non-linear dependencies in a principle way without going through density estimation, it is successfully used for various applications including feature selection [53, 11, 47, 24]. To accurately measure the dependencies from a few labeled instances, the kernel functions need to be suitably designed.

With the proposed method, the kernel functions are modeled by permutation-invariant neural networks [56] that take a small number of labeled instances, called a support set, as input. By this modeling, our model can encode the information of the support set to the kernels, and the kernels are meta-learned with labeled instances in source tasks so that our model can accurately select relevant features even from a few labeled instances. With such kernels, feature selection problems with the support set can be formulated as non-negative least square problems with an $\ell_1$ regularizer, which is introduced in a representative feature selection method, HSIC Lasso [53]. To obtain the global optimal solution (coefficient parameters) in this optimization problem, we use the Iterative Shrinkage Thresholding Algorithm (ISTA) [8], in which each update step can be obtained as a closed-form and is differentiable. This differentiability enables us to efficiently backpropagate the expected test feature selection error in terms of HSIC, which is calculated with testing instances (a query set) drawn from the same task as the support set, after adapting to the support set with gradient-based optimization methods. The proposed method further treats initial coefficient parameters in the optimization problems as trainable parameters, and they are also meta-learned such that only a few update steps by ISTA lead to appropriate coefficient parameters for each task. Since our neural network and initial coefficient parameters are shared across all tasks, the learned model can be applied to unseen target tasks. As a result, the learned model can accurately select features from a small number of labeled instances in a target task.

Our main contributions are summarized as follows: (1) To the best of our knowledge, our work is the first attempt at few-shot learning for supervised feature selection. (2) We propose a novel method

that combines neural networks with HSIC Lasso in the meta-learning framework. Our formulation enables us to perform accurately few-shot feature selection with the closed-from update steps, which is preferable for the efficient meta-learning. (3) We empirically show that the proposed method outperformed existing feature selection methods when there are no sufficient target data in both synthetic and real-word datasets.

## 2   Related Work

Many supervised feature selection methods have been proposed [7, 6]. Least Absolute Shrinkage and Selection Operator (Lasso) is a representative method that uses a sparse linear model for feature selection [48]. Although this method is useful, it cannot find a non-linear relationship between feature and response. To perform non-linear feature selection, several kernel-based methods have been proposed [53, 47, 42]. Among them, HSIC Lasso has performed excellently despite its simple formulation [53, 18, 11, 24]. We use this formulation in our framework since it enables effective and efficient support set adaptation to be performed as described later. Neural network-based feature selection methods have recently been proposed [54, 29, 3, 33]. With the high expressive capability of neural networks, they can find non-linear complex interaction between feature and response. However, they cannot perform well due to overfitting when there are insufficient instances [25, 31]. Although our model also uses neural networks in kernel functions, they are shared across all tasks and are learned with all instances in source tasks. Thus, the proposed method can accurately select relevant features without overfitting.

Multi-task and transfer feature selection methods improve feature selection performances by sharing instances in several tasks [37, 57, 2]. They require instances in target tasks at the training phase. In contrast, the proposed method does not. Our method can quickly adapt to unseen target tasks given target labeled instances at the testing phase.

Meta-learning has recently been attracting a lot of attention, which trains a model such that it generalizes well after adapting to few instances (support set) [17, 50, 46, 44, 19]. Although many methods have been proposed, existing methods are not designed for few-shot supervised feature selection and thus they cannot be applied to our problems. Some studies use the term "meta-learning" for selecting a feature selection algorithm taking meta-features such as number of classes and number of features [45, 22, 39]. They are not methods for few-shot learning, and thus, the scope of these studies is different to that of our study. One method uses the meta-learning framework to select features in hidden features obtained through CNNs for stable prediction [55]. This method cannot perform task-specific feature selection or feature selection from a few data. The gradient-based meta-learning methods such as model-agnostic meta-learning (MAML) [9] adapt to the support set by iteratively updating the whole neural network parameters with gradient descent methods. These methods require second-order derivatives of the parameters of the whole neural network when backpropagating the loss, which imposes considerable computational and memory burdens [4, 43]. Although the proposed method also iteratively adapts to the support set by ISTA, our formulation can adapt to support sets by updating only coefficient parameters for ISTA whose dimension is equal to the feature vector size, and its update step can be obtained as a closed-form. This closed-form update step enables calculation of the second-order derivative of the parameters to be evaded in an automatic differentiation framework such as Pytorch [40], which realizes more efficient adaptation. Encoder-decoder meta-learning such as neural processes [12, 13, 19] efficiently adapts to a support set by forwarding the support set to neural networks. They use only neural networks, and do not explicitly minimize the loss of the support set when adapting to new tasks. Thus, they might have difficulty performing effective adaptation. In contrast, the proposed method explicitly minimize the loss of the support set by solving the optimization problem for feature selection, which achieves accurate adaptation. One few-shot learning method for unsupervised feature selection has recently been proposed that selects features from few unlabeled instances [25]. Unlike our method, it cannot use labels. This method is categorized as an encoder-decoder method, which selects features with only neural networks. Thus, it might be difficult to perform effective adaptation. We show that our method outperforms this method in our experiments. Some methods use deep kernels, which combine neural networks with kernels, in the meta-learning framework [41, 20, 49]. They are designed for classification or regression problems, and cannot be used for feature selection. We design novel kernel functions with neural networks that are appropriate for feature selection, which are described in Section 4.

## 3 Preliminary

In this section, we briefly review HSIC, which can measure non-linear dependencies between two random variables without going through density estimation. Let $X$ and $Y$ be two random variables on domains $\mathcal{X}$ and $\mathcal{Y}$, respectively, and sample $(x, y)$ is drawn from a joint distribution $p(x, y)$. HSIC is given by

$$
\begin{aligned}
\text{HSIC}(X, Y) =& \mathbb{E}_{x,x',y,y'}[K(x, x')L(y, y')] + \mathbb{E}_{x,x'}[K(x, x')]\mathbb{E}_{y,y'}[L(y, y')] \\
& - 2\mathbb{E}_{x,y}\left[\mathbb{E}_{x'}[K(x, x')]\mathbb{E}_{y'}[L(y, y')]\right] \geq 0,
\end{aligned} \tag{1}
$$

where $K : \mathcal{X} \times \mathcal{X} \to \mathbb{R}$ and $L : \mathcal{Y} \times \mathcal{Y} \to \mathbb{R}$ are positive-define kernels, $x'(y')$ is a i.i.d. copy of $x(y)$, and $\mathbb{E}_{x,x',y,y'}$ means the expectation over independent pairs $(x, y)$ and $(x', y')$ drawn from $p(x, y)$. When kernels are characteristic such as RBF kernels, HSIC takes zero if and only if $X$ and $Y$ are statistically independent [15, 47]. In other words, a large value of HSIC indicates that there is a strong dependency between $X$ and $Y$. Therefore, HSIC can be used as a dependence measure.

## 4 Proposed Method

### 4.1 Problem Formulation

Let $\mathcal{D}_t := \{\mathbf{x}_{tn}, y_{tn}\}_{n=1}^{N_t}$ be a set of labeled instances in the $t$-th task, where $\mathbf{x}_{tn} = (x_{tn}^{(1)}, \ldots, x_{tn}^{(D)})^\top$ is the $D$-dimensional feature vector of the $n$-th instance in the $t$-th task and $y_{tn}$ is its response. The proposed method can be applied in both regression and classification settings. In this section, for simplicity, we explain our method with regression problems, i.e., $y_{tn} \in \mathbb{R}$. Our model for classification is described in the supplemental material (Section B). We assume that each task has the same feature vector size $D$, but each joint distribution $p_t(\mathbf{x}, y)$ can differ, which is the standard assumption used in transfer learning studies [38]. Suppose that $T$ source tasks $\mathcal{D} := \{\mathcal{D}_t\}_{t=1}^T$ are given at the training phase. At the testing phase, we are given a small number of labeled instances $\mathcal{S} := \{\mathbf{x}_n, y_n\}_{n=1}^{N_\mathcal{S}}$, called a support set, in a target task, which is different from source tasks. Our goal is to select features that are appropriate for the target task from $\mathcal{S}$.

### 4.2 Model

We explain our model that outputs the importance of each feature given $\mathcal{S}$. By selecting the top-$K$ features in accordance with the importance values, we can select $K$ relevant features.

**Optimization Problem**  To estimate importance values $\boldsymbol{\alpha} = (\alpha_1, \ldots, \alpha_D) \in \mathbb{R}_{\geq 0}^D$, we consider the following optimization problem with support set $\mathcal{S}$:

$$
\min_{\boldsymbol{\alpha} \in \mathbb{R}_{\geq 0}^D} \frac{1}{2N_\mathcal{S}^2} \| \bar{\mathbf{L}}_\mathcal{S} - \sum_{d=1}^D \alpha_d \bar{\mathbf{K}}_\mathcal{S}^{(d)} \|_F^2 + \lambda \| \boldsymbol{\alpha} \|_1, \tag{2}
$$

where $\| \cdot \|_F$ is the Frobenius norm, $\| \cdot \|_1$ is the $\ell_1$ norm, $\lambda \in \mathbb{R}_{>0}$ is a regularization parameter, $\bar{\mathbf{L}}_\mathcal{S} := \boldsymbol{\Gamma}_{N_\mathcal{S}} \mathbf{L}_\mathcal{S} \boldsymbol{\Gamma}_{N_\mathcal{S}} \in \mathbb{R}^{N_\mathcal{S} \times N_\mathcal{S}}$ is the centered Gram matrix, $(\mathbf{L}_\mathcal{S})_{ij} := L(\mathcal{S})(y_i, y_j) \in \mathbb{R}$ is the $\mathcal{S}$-dependent kernel for the response in $\mathcal{S}$, $\boldsymbol{\Gamma}_{N_\mathcal{S}} := \mathbf{I}_{N_\mathcal{S}} - \frac{1}{N_\mathcal{S}} \mathbf{1}_{N_\mathcal{S}} \mathbf{1}_{N_\mathcal{S}}^\top$ is the centering matrix, $\mathbf{I}_{N_\mathcal{S}}$ is the $N_\mathcal{S}$-dimension identity matrix, $\mathbf{1}_{N_\mathcal{S}}$ is the $N_\mathcal{S}$-dimension vector whose elements are all one, $\bar{\mathbf{K}}_\mathcal{S}^{(d)} := \boldsymbol{\Gamma}_{N_\mathcal{S}} \mathbf{K}_\mathcal{S}^{(d)} \boldsymbol{\Gamma}_{N_\mathcal{S}} \in \mathbb{R}^{N_\mathcal{S} \times N_\mathcal{S}}$ is the centered Gram matrix for the $d$-th feature, and $(\mathbf{K}_\mathcal{S}^{(d)})_{ij} := K^{(d)}(\mathcal{S})(x_i^{(d)}, x_j^{(d)}) \in \mathbb{R}$ is the $\mathcal{S}$-dependent kernel for the $d$-th feature in $\mathcal{S}$. Since kernels $K^{(d)}(\mathcal{S})$ and $L(\mathcal{S})$ depend on support set $\mathcal{S}$, we can reflect information of the support set to the kernels so that our model can accurately select features from the support set. The specific form of the kernels will be explained later.

We explain the interpretation of the problem (2). The first term in Eq. (2) can be rewritten as

$$
\frac{1}{2N_\mathcal{S}^2} \| \bar{\mathbf{L}}_\mathcal{S} - \sum_{d=1}^D \alpha_d \bar{\mathbf{K}}_\mathcal{S}^{(d)} \|_F^2 = - \sum_{d=1}^D \alpha_d \widehat{\text{HSIC}}_\mathcal{S}(X^{(d)}, Y) + \frac{1}{2} \sum_{d,d'=1}^D \alpha_d \alpha_{d'} \widehat{\text{HSIC}}_\mathcal{S}(X^{(d)}, X^{(d')}) + C, \tag{3}
$$

where $\widehat{\mathrm{HSIC}}_{\mathcal{S}}(X^{(d)}, Y) := \frac{1}{N_{\mathcal{S}}^2}\mathrm{Tr}(\bar{\mathbf{K}}_{\mathcal{S}}^{(d)}\bar{\mathbf{L}}_{\mathcal{S}})$ is the empirical estimate of HSIC with support set $\mathcal{S}$ [15], $\mathrm{Tr}(\cdot)$ is the trace, and $C$ is a constant term. If there is a strong dependency between the $d$-th feature $X^{(d)}$ and response $Y$, $\widehat{\mathrm{HSIC}}_{\mathcal{S}}(X^{(d)}, Y)$ takes a large value and $\alpha_d$ should also take a large value to minimize Eq. (2). If $X^{(d)}$ is independent of $Y$, $\widehat{\mathrm{HSIC}}_{\mathcal{S}}(X^{(d)}, Y)$ is close to zero and thus $\alpha_d$ tends to become zero by the $\ell_1$ regularizer. Thus, we can regard $\alpha_d$ as the importance of the $d$-th feature. Moreover, if $X^{(d)}$ and $X^{(d')}$ are strongly dependent, i.e., redundant features, $\widehat{\mathrm{HSIC}}_{\mathcal{S}}(X^{(d)}, X^{(d')})$ takes a large value and thus, either of $\alpha_d$ or $\alpha_d'$ tends to zero. This means that redundant features are automatically removed. Therefore, by minimizing Eq. (2), we can select relevant features for predicting the response while removing redundant features. However, when there are insufficient instances, the empirical estimate of HSIC will become inaccurate, and thus, relevant features will be difficult to accurately select.

**Support Set-dependent Kernel Functions** To alleviate this problem, we propose kernel functions appropriate for meta-learning feature selection with a small number of labeled instances. First, we consider obtaining a latent representation of a task from support set $\mathcal{S}$ by using permutation-invariant neural networks [56]:

$$\mathbf{z} := g\left(\frac{1}{N_{\mathcal{S}}}\sum_{(\mathbf{x}, y)\in\mathcal{S}} f([\mathbf{x}, y])\right) \in \mathbb{R}^J, \tag{4}$$

where $f$ and $g$ are any feed-forward neural network, and $[\cdot, \cdot]$ is concatenation. Since averaging operation is permutation-invariant, the neural network in Eq. (4) outputs the same vector even when the order of instances in the support set varies. Moreover, this neural network can handle different numbers of instances. Thus, the neural network in Eq. (4) is well defined as functions for set inputs. Latent task representation $\mathbf{z}$ contains information of the empirical distribution of instances in support set $\mathcal{S}$. Although we use averaging operation for simplicity, we can use any other permutation-invariant function such as summation, max [56], and set transformer [28] to obtain latent task representations.

By using latent task representation $\mathbf{z}$, our kernel functions are defined by

$$K^{(d)}(\mathcal{S})(x_i^{(d)}, x_j^{(d)}) := \beta^{(d)}(\mathbf{z})\exp\left(-\sigma^{(d)}(\mathbf{z})(x_i^{(d)} - x_j^{(d)})^2\right),$$

$$L(\mathcal{S})(y_i, y_j) := \exp\left(-\frac{1}{2\sigma_{\mathrm{y}}^2}(y_i - y_j)^2\right), \tag{5}$$

where $\boldsymbol{\beta}(\mathbf{z}) := (\beta^{(1)}(\mathbf{z}), \ldots, \beta^{(D)}(\mathbf{z})) \in \mathbb{R}_{>0}^D$ and $\boldsymbol{\sigma}(\mathbf{z}) := (\sigma^{(1)}(\mathbf{z}), \ldots, \sigma^{(D)}(\mathbf{z})) \in \mathbb{R}_{>0}^D$ are modeled by feed-forward neural networks taking $\mathbf{z}$ as input, respectively, and $\sigma_{\mathrm{y}} > 0$ is a hyperparameter. We use a $\mathcal{S}$-independent RBF kernel for $y$ for simplicity although any $\mathcal{S}$-dependent kernel can be used. $\beta^{(d)}(\mathbf{z})$ and $\sigma^{(d)}(\mathbf{z})$ can vary in accordance with $\mathbf{z}$. Intuitively, we expect to obtain *inductive bias* appropriate for feature selection in each task, i.e., each feature is easily selected or it is not selected, by meta-learning $\boldsymbol{\beta}(\mathbf{z})$ and $\boldsymbol{\sigma}(\mathbf{z})$. Specifically, for example, when $\beta^{(d)}(\mathbf{z})$ is learned to zero, both $\widehat{\mathrm{HSIC}}_{\mathcal{S}}(X^{(d)}, Y)$ and $\widehat{\mathrm{HSIC}}_{\mathcal{S}}(X^{(d)}, X^{(d')})$ become zero in Eq. (3). Thus, corresponding $\alpha_d$ tends to be made zero by the $\ell_1$ regularizer. Similarly, $\sigma^{(d)}(\mathbf{z})$ can be learned to fit the scale of the feature in each task. These kernel parameters are meta-learned such that our model can accurately select relevant features from a small number of labeled instances, which will be explained in detail in the next subsection. We note that the kernels in Eq. (5) are characteristic if $\beta^{(d)}(\mathbf{z})$ and $\sigma^{(d)}(\mathbf{z})$ are bounded since RBF kernel is characteristic.

**Support Set Adaptation by ISTA** To solve the optimization problem (2), we use ISTA, which is a simple and efficient algorithm for solving $\ell_1$-regularized problems in a mathematically elegant way [8]. By using ISTA, we can derive the closed-form update steps for optimization. Specifically, we can solve the problem (2) by repeating the following steps:

$$\boldsymbol{\alpha} \leftarrow \boldsymbol{\alpha} - \frac{\mu}{N_{\mathcal{S}}^2}\mathbf{A}^\top(\mathbf{A}\boldsymbol{\alpha} - \mathbf{b}), \tag{6}$$

$$\boldsymbol{\alpha} \leftarrow \mathrm{R}_{\lambda\mu}(\boldsymbol{\alpha}) := [\boldsymbol{\alpha} - \lambda\mu\mathbf{1}_D]_+, \tag{7}$$

where $\mathbf{A} := \left[\mathrm{vec}(\bar{\mathbf{K}}_{\mathcal{S}}^{(1)}), \ldots, \mathrm{vec}(\bar{\mathbf{K}}_{\mathcal{S}}^{(D)})\right] \in \mathbb{R}^{N_{\mathcal{S}}^2 \times D}$, $\mathbf{b} := \mathrm{vec}(\bar{\mathbf{L}}_{\mathcal{S}}) \in \mathbb{R}^{N_{\mathcal{S}}^2}$, where $\mathrm{vec}(\cdot)$ is the vectorization operator, $\mathrm{R}_{\lambda\mu}$ is a non-negative soft-thresholding operator, and $[\cdot]_+ = \mathrm{ReLU}(\cdot)$. The

---

**Algorithm 1** Training procedure of our model.

---

**Require:** Datasets in source tasks $\mathcal{D}$, support set size $N_{\mathcal{S}}$, query set size $N_{\mathcal{Q}}$, step size for ISTA $\mu$, and the number of iterations for ISTA $I$
**Ensure:** Parameters of our model $\Theta$
 1: **repeat**
 2:     Randomly sample task $t$ from $\{1, \ldots, T\}$
 3:     Randomly sample support set $\mathcal{S}$ with size $N_{\mathcal{S}}$ from $\mathcal{D}_t$
 4:     Randomly sample query set $\mathcal{Q}$ with size $N_{\mathcal{Q}}$ from $\mathcal{D}_t \setminus \mathcal{S}$
 5:     Calculate latent task representation $\mathbf{z}$ from support set $\mathcal{S}$ by Eq. (4)
 6:     Calculate kernel matrices (5) with task representation $\mathbf{z}$ and support set $\mathcal{S}$
 7:     **for** $l := 1$ to $I$ **do**
 8:         Update coefficient parameters (importance values) $\boldsymbol{\alpha}$ by Eqs. (6) and (7)
 9:     **end for**
10:     Calculate the loss $J(\mathcal{Q}; \mathcal{S})$ on query set $\mathcal{Q}$ by Eqs. (9) and (10)
11:     Update parameters $\Theta$ with the gradients of the loss $J(\mathcal{Q}; \mathcal{S})$
12: **until** End condition is satisfied;

---

first step (6) minimizes the first term in Eq. (2) by using gradient descent with step size $\mu > 0$. The second step (7) reflects the second term in Eq. (2) and non-negative constants. It shrinks $\boldsymbol{\alpha}$ towards zero while satisfying non-negative constraints. We treat the initial coefficient parameters (importance values) $\boldsymbol{\alpha}_0$ for ISTA as trainable parameters like the MAML [9]. By learning good initial parameters with multiple tasks, we can obtain appropriate parameters for each target task even with a few iterations of ISTA.

### 4.3 Training

We explain our training procedure. In this subsection, symbol $\mathcal{S}$ is used as a support set in source tasks. In our model, the parameters to be estimated, $\Theta$, are parameters of neural networks $f$, $g$, $\boldsymbol{\beta}$, $\boldsymbol{\sigma}$, initial coefficient parameters $\boldsymbol{\alpha}_0$, and regularization parameter $\lambda$. We estimate these parameters by minimizing the expected test error on features selected from support set $\mathcal{S}$, where support set and test instances $\mathcal{Q}$, called the query set, are randomly generated from source tasks:

$$\mathbb{E}_{t \sim \{1, \ldots, T\}} \left[ \mathbb{E}_{(\mathcal{S}, \mathcal{Q}) \sim \mathcal{D}_t} \left[ J(\mathcal{Q}; \mathcal{S}) \right] \right], \tag{8}$$

where

$$J(\mathcal{Q}; \mathcal{S}) := \frac{1}{2N_{\mathcal{Q}}^2} \| \bar{\mathbf{L}}_{\mathcal{Q}} - \sum_{d=1}^{D} \alpha_d^* \bar{\mathbf{K}}_{\mathcal{Q}}^{(d)} \|_{\mathrm{F}}^2, \tag{9}$$

where $\boldsymbol{\alpha}^* = (\alpha_1^*, \ldots, \alpha_D^*) \in \mathbb{R}_{\geq 0}^D$ is coefficient parameters obtained from $\mathcal{S}$ by ISTA, $\bar{\mathbf{L}}_{\mathcal{Q}} := \boldsymbol{\Gamma}_{N_{\mathcal{Q}}} \mathbf{L} \boldsymbol{\Gamma}_{N_{\mathcal{Q}}} \in \mathbb{R}^{N_{\mathcal{Q}} \times N_{\mathcal{Q}}}$ is the Gram matrix for the response on query set $\mathcal{Q}$ with size $N_{\mathcal{Q}}$, $\mathbf{L}_{ij} := L(y_i, y_j) \in \mathbb{R}$ is the kernel for the response in $\mathcal{Q}$, $\bar{\mathbf{K}}_{\mathcal{Q}}^{(d)} := \boldsymbol{\Gamma}_{N_{\mathcal{Q}}} \mathbf{K}^{(d)} \boldsymbol{\Gamma}_{N_{\mathcal{Q}}} \in \mathbb{R}^{N_{\mathcal{Q}} \times N_{\mathcal{Q}}}$ is the Gram matrix for the $d$-th feature on $\mathcal{Q}$, and $\mathbf{K}_{ij}^{(d)} := K^{(d)}(x_i^{(d)}, x_j^{(d)}) \in \mathbb{R}$ is the kernel for the $d$-th feature in $\mathcal{Q}$. When query size $N_{\mathcal{Q}}$ is sufficiently large, the empirical estimate of HSIC with the RBF kernels becomes accurate [15]. Thus, we used the following RBF kernels for all features in Eq. (9) as in [53],

$$K^{(d)}(x_i^{(d)}, x_j^{(d)}) := \exp\left( -\frac{1}{2\sigma_{\mathrm{x}}^2} (x_i^{(d)} - x_j^{(d)})^2 \right), \tag{10}$$

where $\sigma_{\mathrm{x}} > 0$ is a hyperparameter. For the kernel of the response $L$, we used the same RBF kernel as that for the support set adaptation in Eq. (5). By using the same kernels for all features, we can treat each feature equally. That is, the proposed method can fairly evaluate the importance of each feature obtained from $\mathcal{S}$ on query set $\mathcal{Q}$.

Algorithm 1 shows the pseudocode for our training procedure. For each iteration, we randomly sample task $t$ from source tasks (Line 2). From $\mathcal{D}_t$, we randomly sample support set $\mathcal{S}$ and query set $\mathcal{Q}$ (Lines 3 and 4). We calculate task representation $\mathbf{z}$ from $\mathcal{S}$ (Line 5), and the $\mathcal{S}$-dependent kernel matrices (Line 6). We iteratively update coefficient parameters $\boldsymbol{\alpha}$ with support set $\mathcal{S}$ by using ISTA (Lines 7 – 9). Using the adapted coefficient parameter $\boldsymbol{\alpha}^*$, we calculate loss $J(\mathcal{Q}; \mathcal{S})$ on query set $\mathcal{Q}$

(Line 10). Lastly, the parameters of our model $\Theta$ are updated with the gradient of the loss (Line 11). Since each update step of ISTA (Eqs. (6) and (7)) is obtained as a closed-form that is differentiable, the whole training can be performed by using gradient-based optimization methods. This training explicitly matches the selected features from few labeled instances $\mathcal{S}$ and those from sufficient labeled instances $\mathcal{Q}$ on the basis of HSIC. Thus, the learned model can select relevant features for each target task given a small number of labeled target instances (specifically, by running lines 5 through 9 of Algorithm 1 with the target instances.)

## 5 Experiments

In this section, we demonstrate the effectiveness of the proposed method using one synthetic and three real-world datasets.

### 5.1 Comparison Methods

We compared the proposed method with seven existing feature selection methods: Lasso, HSIC Lasso (HLasso), feature selection with stochastic gate (STG) [54], STG-S, fine-tuning variant of STG (STG-FT), meta-learning method for feature selection based on Gumbel-softmax distribution (MetaGS) [25], and a supervised variant of MetaGS (SMetaGS). Lasso is a well-known linear feature selection method. HLasso is a kernel-based non-linear feature selection method. STG is a neural network-based non-linear feature selection method that has been reported to outperform various existing methods in various datasets [54]. STG learns a mask vector that determines which features are selected while simultaneously learning predictors. Lasso, HLasso, and STG uses the target support set for feature selection. STG-S trains the model of STG with labeled instances in source tasks. STG-FT fine-tunes the model trained by STG-S with the target support set. MetaGS is a neural network-based encoder-decoder meta-learning method for unsupervised feature selection that uses the reconstruction error for feature selection. SMetaGS is a supervised extension of MetaGS that uses the cross-entropy loss as the objective function. (S)MetaGS meta-learns the mask vector for feature selection. Specifically, the task-specific mask is modeled by the permutation-invariant neural network that takes support set (a few data) as input and is meta-learned to improve test feature selection performance. All methods except for MetaGS use label information. The proposed method, STG-FT, MetaGS, and SMetaGS use both the target support set and source instances for feature selection.

### 5.2 Simulation Experiments

**Data** We first used simple synthetic regression problems with 100-dimensional feature and one-dimensional response spaces. Response $y$ in the $t$-th task was generated from the following non-linear function: $y = w_{t1}x^{(i_1)}\exp(2w_{t2}x^{(i_2)}) + w_{t3}(x^{(i_3)})^2$, where task-specific parameters $w_{t1}$, $w_{t2}$, and $w_{t3}$ were uniform randomly generated from $[-1, 1]$, and features $(x^{(1)}, \ldots, x^{(100)})$ was generated from 100-dimensional standard Gaussian distribution. Feature indexes $(i_1, i_2, i_3)$ represent important feature indexes in the $t$-th task, and were uniform randomly selected for each task from 10 predefined feature indexes $\{(1, 2, 3), (4, 5, 6), \ldots, (28, 29, 30)\}$. We generated 300 labeled instances for each task. We standardized the values of response. We randomly created 300 training, 20 validation, and 100 target tasks.

**Settings** We evaluated the average fraction of correctly selected features on each target task with different numbers of target support instances within $\{10, 20, 30\}$. For each method, we selected the top-3 features by ranking their importance scores. For the proposed method, we selected the hyperparameter on the basis of mean validation loss. For comparison methods, the best test results are reported from their hyperparameter candidates. The details of the experimental settings such as hyperparameter candidates are described in the supplemental material (Section C).

**Results** Table 1 shows average and standard errors of the fraction of correctly selected features with different numbers of target support instances. The proposed method clearly outperformed the other methods. HLasso outperformed Lasso since it is the non-linear method that fits on the non-linear synthetic data. Although STG is also a non-linear method, it did not perform well since training data was too small to train its neural networks. STG-FT, which is the fine-tuning method, improved performance compared with STG-S. Although MetaGS and SMetaGS are meta-learning methods like

Table 1: Averages and standard errors of fraction of correctly selected features [%] with different numbers of target support instances $N_\mathcal{S}$ on the synthetic dataset.

| $N_\mathcal{S}$ | Ours | Lasso | HLasso | STG | STG-S | STG-FT | MetaGS | SMetaGS |
|---|---|---|---|---|---|---|---|---|
| 10 | **39.3±1.9** | 22.7±2.3 | 23.7±1.3 | 11.0±1.3 | 11.7±1.6 | 14.7±2.1 | 7.3±1.4 | 3.7±1.0 |
| 20 | **49.7±1.7** | 30.0±2.1 | 43.7±1.7 | 12.7±1.2 | 11.7±1.6 | 23.0±2.3 | 9.3±1.5 | 7.3±1.4 |
| 30 | **58.3±2.0** | 33.0±2.2 | 48.3±1.8 | 15.0±1.2 | 11.7±1.6 | 24.7±2.3 | 4.0±1.1 | 4.0±1.1 |
| Avg. | **49.1±1.2** | 28.6±1.3 | 38.6±1.2 | 12.9±0.9 | 11.7±0.9 | 20.8±1.3 | 6.9±0.8 | 5.0±0.7 |

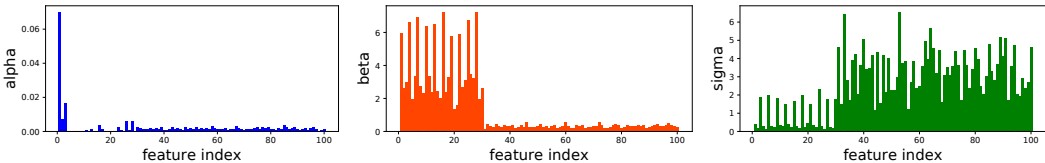

Figure 2: One example of feature importance $\boldsymbol{\alpha}^*$ and kernel parameters, $\boldsymbol{\beta}(\mathbf{z})$ and $\boldsymbol{\sigma}(\mathbf{z})$, obtained by the proposed method in the synthetic dataset with $N_\mathcal{S} = 20$. The figures are arranged from left to right in the order $\boldsymbol{\alpha}^*$, $\boldsymbol{\beta}(\mathbf{z})$, and $\boldsymbol{\sigma}(\mathbf{z})$. True important feature indexes on this example were 1, 2, and 3. Our method accurately assigned the top 3 maximum values of $\alpha_d$ to the true features.

the proposed method, it performed worse. This result suggests that task-specific features are difficult to select from few instances only by using neural networks. The proposed method can select features by explicitly solving Eq. (2) with useful knowledge in source tasks, and thus, it worked well.

Figure 2 shows one example of feature importance $\boldsymbol{\alpha}^*$ and kernel parameters, $\boldsymbol{\beta}(\mathbf{z})$ and $\boldsymbol{\sigma}(\mathbf{z})$, obtained by the proposed method. As explained in Section 4, $\beta^d(\mathbf{z})$ is expected to indicate how likely the $d$-th feature is to be selected before solving Eq. (2). That is, when $\beta^d(\mathbf{z})$ is small, the $d$-th feature is less likely to be selected. $\beta^d(\mathbf{z})$ took a small value when $d$ was in $\{31, \ldots, 100\}$. Since there are no important features within $\{31, \ldots, 100\}$ in all tasks of this synthetic data as described above, $\boldsymbol{\beta}(\mathbf{z})$ was learned as expected. Similarly, $\sigma^d(\mathbf{z})$ tends to take different values in the two ranges $\{1, \ldots, 30\}$ and $\{31, \ldots, 100\}$. This result means that appropriate scales were automatically learned for feature groups that are easy to select (i.e., $d \in \{1, \ldots, 30\}$) and those that are not (i.e., $d \in \{31, \ldots, 100\}$). After adaptation to the support set, $\alpha_d^*$ took a large value when $d$ was in $\{1, 2, 3\}$, which is the true important feature indexes of this example. This result shows that the proposed method worked well as expected. We also confirmed that the proposed method works well on more difficult synthetic data in the supplemental material (Section D.10).

### 5.3 Real-world Data Experiments

**Data** We used there real-world datasets: Mnistr[1], Isolet[2], and IoT[3], which have been widely used in previous studies [3, 25, 54]. Mnistr is derived from Mnist by rotating the images [14]. This dataset has six tasks (six rotation angles), and its feature dimension is 256. We created the binary classification problem by regarding even digits as positive and odd digits as negative, respectively. Isolet consists of letters (classes) spoken by 150 speakers, and speakers are grouped into five groups (tasks) by speaking similarity. Each instance is represented as a 617-dimensional vector. We created the binary classification problem by regarding class labels $\{1, \ldots, 13\}$ as positive and $\{14, \ldots, 26\}$ as negative, respectively. IoT is real network traffic data, which are gathered from nine IoT devices (tasks) infected by malware [35]. Each task has normal and malicious traffic instances, and each instance is represented by a 115-dimensional vector. For each task, we randomly used 500 normal and 500 malicious instances. For all datasets, we standardized the values of each response and feature. We randomly select one task for the target task, one task for the validation task, and the rest for the source tasks. For each dataset, we created 10 different target/validation/source task splits.

**Settings** To evaluate the quality of selected features, we trained classifiers by using the target support set with the selected features and evaluated the test classification accuracy, which is the

---

[1] https://github.com/ghif/mtae

[2] http://archive.ics.uci.edu/ml/datasets/ISOLET

[3] https://archive.ics.uci.edu/ml/datasets/detection_of_IoT_botnet_attacks_N_BaIoT

Table 2: Averages and standard errors of test accuracy [%] with different numbers of target support instances on the real-world datasets.

| Data | $N_S$ | Ours | Lasso | HLasso | STG | STG-S | STG-FT | MetaGS | SMetaGS |
|---|---|---|---|---|---|---|---|---|---|
| Mnistr | 10 | 67.1±0.9 | 60.8±0.8 | 62.8±0.7 | 61.1±0.9 | 67.7±0.9 | **67.8±1.0** | 59.3±1.0 | 64.4±0.9 |
| | 20 | **72.0±0.7** | 65.1±0.7 | 67.9±0.8 | 67.9±0.7 | 70.9±1.0 | 70.6±1.1 | 63.6±0.9 | 68.4±1.3 |
| | 30 | **75.2±0.5** | 68.5±1.0 | 71.8±0.7 | 70.3±1.0 | 73.4±1.0 | 73.4±1.0 | 69.2±1.2 | 73.0±1.0 |
| Avg. | | **71.4±0.5** | 64.8±0.6 | 67.5±0.6 | 66.5±0.7 | 70.7±0.6 | 70.6±0.6 | 64.0±0.7 | 68.6±0.7 |
| Isolet | 10 | **67.3±0.6** | 58.1±1.0 | 58.1±0.8 | 59.2±0.7 | 57.9±1.0 | 58.3±1.0 | 58.7±0.8 | 61.7±0.8 |
| | 20 | **67.3±0.7** | 63.3±0.7 | 61.5±0.8 | 62.0±0.7 | 66.4±0.9 | **67.3±0.9** | 60.2±0.9 | 66.5±0.8 |
| | 30 | **69.4±0.5** | 64.2±0.6 | 61.4±0.6 | 63.5±0.7 | 65.5±1.2 | 65.9±1.2 | 63.6±0.7 | 69.0±1.0 |
| Avg. | | **68.0±0.4** | 61.9±0.5 | 60.3±0.5 | 61.6±0.4 | 63.3±0.7 | 63.9±0.7 | 60.8±0.6 | 65.7±0.6 |
| IoT | 10 | **71.3±1.3** | 68.9±0.1 | 69.0±0.1 | 69.0±0.1 | 68.7±0.2 | 69.5±0.8 | 71.1±1.0 | 69.5±0.2 |
| | 20 | 72.3±1.6 | 71.9±1.7 | 68.7±0.1 | 68.7±0.2 | **74.0±2.0** | **74.0±2.0** | 69.3±0.2 | 71.0±1.3 |
| | 30 | **74.3±2.0** | 73.4±2.1 | 68.4±0.1 | 68.4±0.1 | 73.9±2.1 | 73.9±2.1 | 70.7±1.2 | 71.8±1.5 |
| Avg. | | **72.6±1.0** | 71.4±0.9 | 68.7±0.1 | 68.7±0.1 | 72.2±1.0 | 72.5±1.0 | 70.4±0.5 | 70.8±0.7 |

Table 3: Ablation studies of our model: averages and standard errors of test accuracy [%] over different numbers of target support instances and different numbers of selected features on each real-world datasets.

| Data | Ours | w/o Latent | w/o Feature | w/o LFeature | w/o $\mathcal{S}$-kernel | w/o Initial | w/ Deep | w/o $\mathcal{S}$-adapt |
|---|---|---|---|---|---|---|---|---|
| Mnistr | **71.4±0.5** | 70.3±0.6 | 70.0±0.7 | 70.1±0.6 | 70.6±0.7 | 69.4±0.7 | 70.6±0.6 | 70.4±0.7 |
| Isolet | **68.0±0.4** | 65.7±0.4 | 66.1±0.4 | 65.8±0.4 | 65.6±0.4 | 65.1±0.4 | 64.0±0.4 | 64.2±0.5 |
| IoT | 72.6±1.0 | **74.0±1.1** | 71.6±0.9 | 70.4±0.7 | 70.6±0.7 | 71.8±0.9 | 70.7±0.7 | 73.4±1.0 |
| Avg. | **70.7±0.4** | 70.0±0.5 | 69.3±0.4 | 68.8±0.4 | 68.9±0.4 | 68.8±0.4 | 68.4±0.4 | 69.3±0.5 |

standard evaluation protocol used in the previous feature selection studies [53, 47, 29, 54]. Since the proposed method, HLasso, and MetaGS cannot directly classify test instances, we used support vector machines with RBF kernels as the classifiers for all methods for a fair comparison. We did not use neural networks as classifiers since they have many hyperparameters to be tuned, and target training instances are too small to train without overfitting. The details of the settings such as hyperparameters are described in the supplemental material (Section C).

**Results**  Table 2 shows average and standard errors of test accuracies over different numbers of selected features within $\{10, 30, 50\}$ when changing the numbers of target support instances within $\{10, 20, 30\}$. The proposed method achieved the best average accuracies on all datasets. Lasso, HLasso, and STG, which used only the target support instances, did not perform well since it was difficult to select relevant features from only few labeled instances. STG-FT outperformed these methods for all datasets, which indicates that using information on related tasks is useful to alleviate the lack of target data. MetaGS did not work well on all datasets, which indicates the effectiveness of label information. SMetaGS did not perform well with Mnistr and IoT since the support set adaptation was difficult with only neural networks. In contrast, the proposed method worked well by appropriately unifying neural networks and kernel methods in a meta-learning framework.

Figure 3 shows average and standard errors of test accuracies when changing the number of selected features $K$ on Isolet. The proposed method performed well across different $K$ values. In particular, when $K$ is small, the proposed method outperformed the others by a large margin. In general, when $K$ is small, it is more important to select informative features to improve performance. Thus, this result suggests that the proposed method was able to assign high importance to informative features.

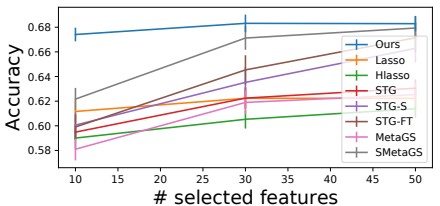

Figure 3: Average and standard errors of test accuracies over different target support instances when changing the selected feature size $K$ on Isolet dataset.

Table 3 shows the results of ablation studies. We compared our model with seven models: w/o Latent, w/o Feature, w/o LFeature, w/o $\mathcal{S}$-kernel, w/o Initial, w/ Deep, and w/o $\mathcal{S}$-adapt. For support set adaptation, w/o Latent uses the kernels without latent task representation **z**, i.e., it uses the task-invariant feature-specific kernel parameters $\beta^{(d)}$ and $\sigma^{(d)}$ in Eq. (5). w/o Feature uses the task-specific feature-invariant

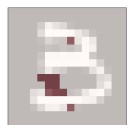 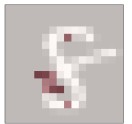 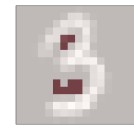 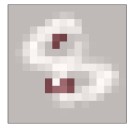 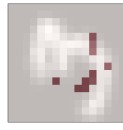 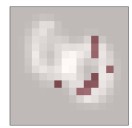

(a) Source 1: 3    (b) Source 1: 8    (c) Source 2: 3    (d) Source 2: 8    (e) Target: 3    (f) Target: 8

Figure 4: Ten selected features on two source and one target tasks (rotation angles) of Mnistr by the proposed method with $N_{\mathcal{S}} = 30$. We compared digits '3' and '8'. Red represents the selected features (pixels).

kernel parameters $\beta(\mathbf{z})$ and $\sigma(\mathbf{z})$ for all features. w/o LFeature uses the task and feature-invariant kernel parameters $\beta$ and $\sigma$ for all features. w/o $\mathcal{S}$-kernel uses the $\mathcal{S}$-independent kernels of Eq. (10). This method adapts to the support set by only learning initial coefficient parameters. w/o Initial uses the same kernels as the proposed method, but does not learn initial coefficient parameters. w/ Deep uses the deep kernels [51], $K^{(d)}(\mathcal{S})(x_i^{(d)}, x_j^{(d)}) := \exp(-(h(x_i^{(d)}, \mathbf{z}) - h(x_j^{(d)}, \mathbf{z}))^2)$, where $h$ is a three-layered feed-forward neural networks with one output node. w/o $\mathcal{S}$-adapt does not perform support set adaptation (i.e., $I = 0$). This method learns task-invariant initial coefficient parameters for feature selection on any tasks. The proposed method showed the best average results over all datasets. Especially, it outperformed w/o Latent, which indicates the usefulness of inferring task-specific kernels for each feature from the support set. In IoT, w/o Latent worked well. This was probably because each task has similar properties as described in [26], and thus, task-invariant kernels might be sufficient. w/o Feature and w/o LFeature performed worse than the proposed method, which indicates that feature-specific kernel parameters are useful to select features. Since w/o $\mathcal{S}$-kernel and w/o Initial did not work well, we found that both the kernel and initial parameter adaptations need to be considered simultaneously. w/ Deep did not perform well. This result indicates that the form of our kernel function in Eq. (5) is essential for our problems. The proposed method also outperformed w/o $\mathcal{S}$-kernel in Mnistr and Isolet, which means that support set adaptation needs to be performed to select task-specific features. Overall, these results indicate the effectiveness of our model design.

We visualized the selected features by the proposed method with Mnistr dataset. Figure 4 shows 10 selected features on each task of Mnistr by the proposed method when $N_{\mathcal{S}} = 30$. Since the selected features differ between tasks, we found that the learned model can select task-specific features by adapting to the support set on each task as expected. In addition, the selected features can identify digits '3' and '8' on each task since they captured the bottom left white pixels of digit '8'. Therefore, the proposed method can identify relevant features that are responsible for the responses.

## 6  Conclusion

In this paper, we proposed a few-shot learning method for feature selection based on Hilbert-Schmidt Independence Criterion (HSIC). Experiments showed that the proposed method outperformed existing feature selection methods when there was insufficient data in target tasks. Although the proposed method is useful as a feature selection method, it cannot train a classification or regression model that can predict unseen instances simultaneously. As future work, we plan to combine HSIC Lasso and neural network-based classification/regression model in our meta-learning framework to simultaneously perform feature selection and classification/regression model learning from a few labeled data.

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
