# Supplemental Material: Few-shot Learning for Feature Selection with Hilbert-Schmidt Independence Criterion

**Atsutoshi Kumagai**
NTT Computer and Data Science Laboratories
atsutoshi.kumagai.ht@hco.ntt.co.jp

**Tomoharu Iwata**
NTT Communication Science Laboratories
tomoharu.iwata.gy@hco.ntt.co.jp

**Yasutoshi Ida**
NTT Computer and Data Science Laboratories
yasutoshi.ida@ieee.org

**Yasuhiro Fujiwara**
NTT Communication Science Laboratories
yasuhiro.fujiwara.kh@hco.ntt.co.jp

## A  Merits of Feature Selection

We explain the merits of feature selection by comparing it to feature extraction. Feature extraction aims to learn instance representations to improve the performance of prediction models such as classification or regression models [9, 4]. The learned representations are represented by the dense combination of all features. Thus, they cannot explain/analyze the dataset itself or reduce the cost of collecting irrelevant or redundant features by identifying the important feature subset from original features, which are the major objectives (merits) of feature selection. Existing meta-learning methods perform feature extraction to improve the classification or regression performance [5, 3, 14, 13]. Therefore, they cannot be used for feature selection purposes. To the best of our knowledge, our work is the first attempt at meta-learning for supervised feature selection, which enables us to select task-specific important feature subsets from a few labeled data.

## B  Proposed Model for Classification Problems

We explain how to apply our model to classification problems. To this end, we slightly modify kernel functions for responses and the way to construct latent task representations described in the main paper.

For the kernel functions for responses $y$ in Eq. (5), following a previous study [15], we can use delta kernels, $L(\mathcal{S})(y_i, y_j) := \frac{1}{|\mathcal{S}_{y_i}|}$ if $y_i = y_j$ and zero otherwise, where $\mathcal{S}_c := \{(\mathbf{x}, y) \in \mathcal{S} | y = c\}$. We can also use this kernel for the objective function with the query set in Eq. (10).

For latent task representation $\mathbf{z}$ in Eq. (4), we change the construction procedure of $\mathbf{z}$ depending on whether the response spaces are the same across tasks or not. When the response space is the same across tasks, we can use the same neural network architecture as that for regression problems by using one-hot vector representations of responses (classes) $y$. When the response spaces are different across tasks, we cannot use the one-hot vector representations for classes because we do not know classes in unseen target tasks at the training phase. To deal with this, for each support set, we first calculate class-specific task representation $\mathbf{z}_c$ where $c$ is a class in the support set,

$$\mathbf{z}_c := g \left( \frac{1}{|\mathcal{S}_c|} \sum_{(\mathbf{x},c) \in \mathcal{S}_c} f(\mathbf{x}) \right),$$

36th Conference on Neural Information Processing Systems (NeurIPS 2022).

and then, we construct $\mathbf{z}$ from the class-specific task representations by using permutation-invariant neural networks,

$$\mathbf{z} := u\left(\frac{1}{C}\sum_c v(\mathbf{z}_c)\right),$$

where $u$ and $v$ are any feed-forward neural networks, and $C$ is the total number of classes in the support set. By the average operation, we can obtain the same vector $\mathbf{z}$ even if the order of the class-specific task representations varies.

## C   Experimental Setting Details

We explain the neural network architectures used in our simulation and real-world data experiments. For the proposed method, we used a three(two)-layered feed-forward neural network for latent task representations $f(g)$ in Eq. (4). The hidden and output sizes of $f$ were 32 and ReLU activations were used. The output size of $g$ was fixed to 4 ($J = 4$). For functions of kernel parameters $\boldsymbol{\beta}$ and $\boldsymbol{\sigma}$ in Eq. (5), two-layered feed-forward neural networks with 32 hidden nodes and ReLU activations were used, respectively. To ensure the non-negativeness of $\boldsymbol{\beta}$ and $\boldsymbol{\sigma}$, softplus functions were used for output nodes. For prediction networks of STG, STG-S, STG-FT, MetaGS, and SMetaGS, we used three-layered neural networks with 32 hidden nodes and ReLU activations, respectively. For MetaGS and SMetaGS, we used the temperature annealing from 10 to 0.01 as in the original paper. All methods were implemented on the basis of Pytorch [11]. All experiments were conducted on a Linux server with an Intel Xeon CPU and a NVIDIA GeForce GTX 1080 GPU.

For the simulation experiments, the hyperparameters of the proposed method was determined on the basis of mean validation loss. Step size for ISTA $\mu$ was selected from $\{10^{-3}, 10^{-2}, 10^{-1}, 1\}$. The number of iterations for ISTA $I$ was selected from $\{1, 3, 10\}$. Gaussian widths $\sigma_x$ and $\sigma_y$ were set to one. For all comparison methods, the best test results are reported from the following hyperparameter candidates. For Lasso, HLasso, STG, STG-T, and STG-FT, the regularization parameter was chosen from $\{10^{-6}, 10^{-5}, \ldots, 10\}$. For HLasso, Gaussian width was chosen from $\{0.1, 0.5, 1, 2\}$. For STG-FT, the number of fine-tune iterations was chosen from $\{10, 50, 100\}$. For all neural network-based methods, we used the Adam optimizer [6] with a learning rate of $10^{-3}$. The mini-batch size was 64 (for the proposed method, we set $N_Q + N_S = 64$). The validation loss was also used for early stopping to avoid over-fitting, where the maximum number of training iterations was 10,000.

For the real-word data experiments, we used the same hyperparameter candidates used in the synthetic dataset. For the proposed method, we used the delta kernel for kernel functions of responses as described in Section B. Since each dataset used in our experiments has the same response (class) space across tasks, we used one-hot vector representations for classes to obtain latent task representations as described in Section B. For all methods, to tune their hyperparameters and the regularization parameter of support vector machine (SVM), we used 10 labeled instances from the target task as validation data, and its validation accuracy was used. We used the scikit-learn implementations of the SVM. The regularization parameter candidates of SVM were $\{10^{-4}, 10^{-3}, \ldots, 10^2\}$. The Gaussian width of the RBF kernel was automatically fitted to training data by scikit-learn.

## D   Additional Experimental Results

### D.1   Test Accuracies with Different Selected Feature Sizes

Figure 1 shows the average and standard errors of test accuracies over different target support instances when changing the selected feature size $K$ on each dataset. The proposed method consistently performed well with different $K$ values on Mnistr and Isolet. In particular, when $K$ is small, the proposed method outperformed the others by a large margin. In general, it is more important to select informative features when $K$ is small. Therefore, this result indicates that the proposed method can select important features accurately. For IoT, the proposed method, STG-S, and STG-FT showed similar results. Especially, these methods performed well when $K$ is small. This would be because this dataset has some noisy features, and thus they deteriorate performance. Overall, the proposed method performed well.

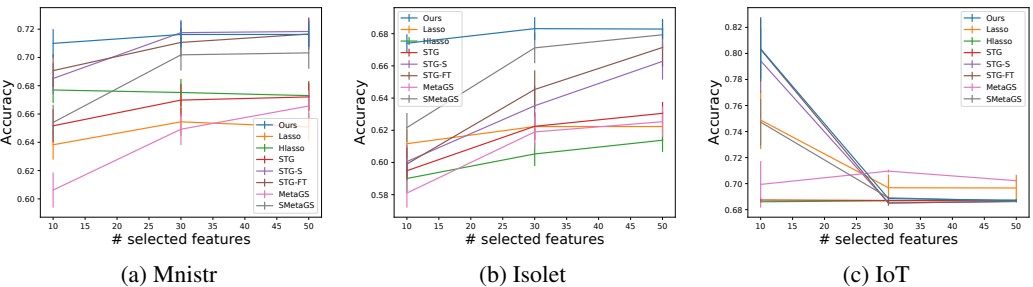

| (a) Mnistr | (b) Isolet | (c) IoT |

Figure 1: Average and standard errors of test accuracies over different target support instances when changing the number of selected features $K$.

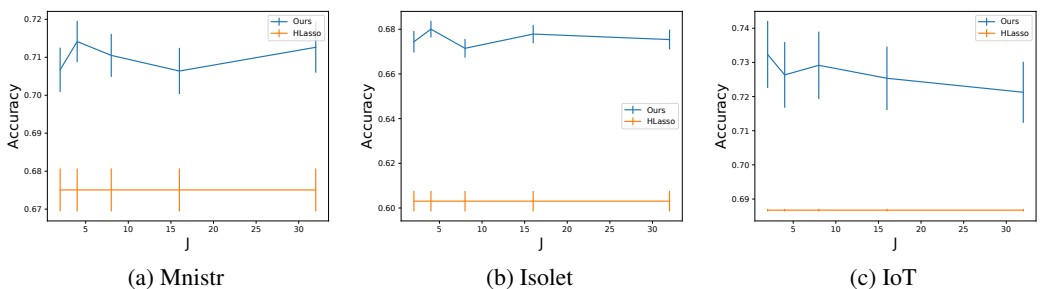

| (a) Mnistr | (b) Isolet | (c) IoT |

Figure 2: Average and standard errors of test accuracies over different target support instance sizes and selected feature sizes when changing the dimension of latent task representations $J$.

### D.2 Dependency of the Dimension of Latent Task Representations

We investigated the dependency of the dimension of latent task representations $\mathbf{z}$ on the proposed method, which are used for inferring the task-specific kernel functions for feature selection. Figure 2 shows the average and standard errors of test accuracies of the proposed method with different dimensions of latent task representations $J$ within $\{2, 4, 8, 16, 32\}$. The proposed method constantly outperformed HLasso and performed relatively stably with different $J$ values.

### D.3 Dependency of the Iteration Numbers of ISTA

We investigated the dependency of the iteration numbers of ISTA on the proposed method. Figure 3 shows the average and standard errors of test accuracies of the proposed method with different iteration numbers $I$ within $\{1, 3, 5, 10, 15\}$. The proposed method consistently outperformed HLasso and performed relatively stably with different $I$ values. This result indicates that the proposed method can select task-specific features with a few training iterations of ISTA.

### D.4 Dependency of the Step Size of ISTA

We investigated the dependency of the step size of ISTA on the proposed method. Figure 4 shows the average and standard errors of test accuracies of the proposed method with different step sizes $\mu$ within $\{10^{-3}, 10^{-2}, 10^{-1}, 1\}$. The proposed method consistently outperformed HLasso with different $\mu$ values. The appropriate step sizes varied across datasets. For Mnistr and Isolet, small step sizes performed well. In contrast, for IoT, large step sizes tended to perform well.

### D.5 Computation Cost

We investigated the computation time of the proposed method. We used a computer with a 2.20 GHz CPU. The target support set size was set to 30 and the iteration number of ISTA was set to 3. Tables 1 and 2 show the training time with source tasks and feature selection time on a target task with Mnistr,

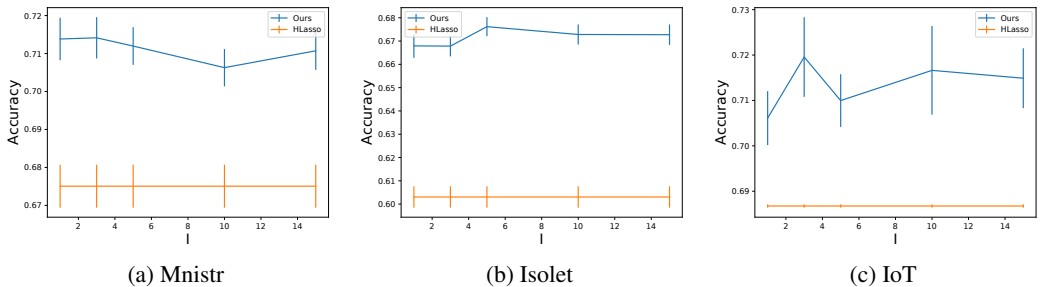

| (a) Mnistr | (b) Isolet | (c) IoT |

Figure 3: Average and standard errors of test accuracies over different target support instance sizes and selected feature sizes when changing the number of iterations of ISTA $I$.

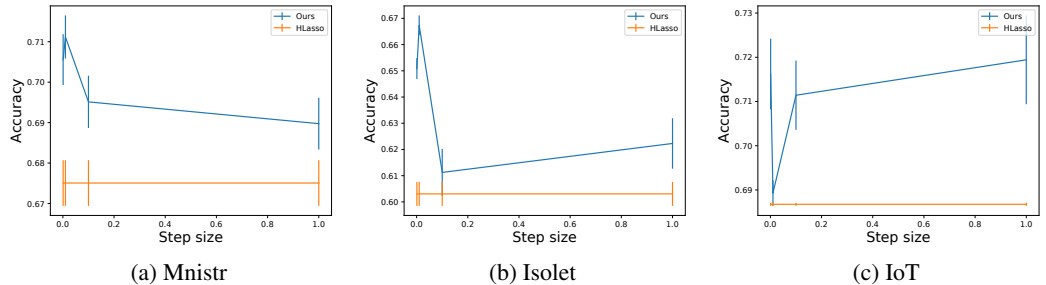

| (a) Mnistr | (b) Isolet | (c) IoT |

Figure 4: Average and standard errors of test accuracies over different target support instance sizes and selected feature sizes when changing the step size of ISTA $\mu$.

Table 1: Training time with source tasks in seconds for each method on Mnistr.

| Ours | STG-S | STG-FT | MetaGS | SMetaGS |
|------|-------|--------|--------|---------|
| 170.7 | 25.1 | 25.1 | 140.2 | 164.4 |

Table 2: Feature selection time on a target task in seconds for each method on Mnistr.

| Ours | Lasso | HLasso | STG | STG-FT | MetaGS | SMetaGS |
|------|-------|--------|-----|--------|--------|---------|
| 0.050 | 0.014 | 1.059 | 22.392 | 0.028 | 0.002 | 0.002 |

Table 3: Detailed analysis of effects of modeling initial coefficient parameters: test accuracy [%] and training time with source tasks in seconds on Mnistr

| Metric | Ours $(I = 3)$ | w/o Initial $(I = 3)$ | w/o Initial $(I = 10)$ | w/o Initial $(I = 20)$ | w/o Initial $(I = 30)$ | w/o Initial $(I = 40)$ | w/o Initial $(I = 50)$ |
|--------|------|-------------|-------------|-------------|-------------|-------------|-------------|
| Train time | 170.7 | 162.5 | 266.2 | 351.9 | 435.3 | 513.7 | 568.9 |
| Test accuracy | 71.4 | 68.6 | 68.3 | 69.0 | 70.3 | 69.0 | 68.3 |

respectively. Since Lasso, HLasso, and STG do not use source tasks, we omitted these methods in Table 1. Since STG-S directly uses selected features from source tasks for the target task, we omitted this method in Table 2. Although the proposed method took time for training with source tasks, it was able to select features on the target task as fast as other methods.

## D.6 Detailed Analysis of Learning Initial Coefficient Parameters

We showed the effectiveness of learning initial coefficient parameters $\alpha_0$ in the ablation study (Table 3 in the main paper). We further analyzed its effect in details. Specifically, we investigated whether w/o Initial, which is our method without learning initial coefficient parameters, can achieve good performance when the number of the iterations for ISTA $I$ is increased. Table 3 shows the mean test accuracy and the training time with source tasks. As the iteration number $I$ was increased, the

Table 4: Averages test accuracy [%] with different numbers of target support instances on the Mnistr with different response (class) spaces.

| $N_S$ | Ours | Lasso | HLasso | STG | STG-S | STG-FT | MetaGS | SMetaGS |
|---|---|---|---|---|---|---|---|---|
| 10 | 59.1 | 54.2 | 58.5 | 58.0 | 56.4 | 59.5 | 57.3 | 57.2 |
| 20 | 65.6 | 60.0 | 62.6 | 63.1 | 64.5 | 61.3 | 59.1 | 59.0 |
| 30 | 68.6 | 61.5 | 67.9 | 66.2 | 65.1 | 68.0 | 64.1 | 63.4 |
| Avg. | 64.4 | 58.6 | 63.0 | 62.4 | 62.0 | 62.9 | 60.2 | 59.9 |

Table 5: Comparison with SVM without feature selection (NoFS): averages of test accuracy [%] over different numbers of target support instances and different numbers of selected features on each real-world dataset.

| Data | Ours | NoFS |
|---|---|---|
| Mnistr | 71.4 | 66.2 |
| Isolet | 68.0 | 65.1 |
| IoT | 72.6 | 68.6 |
| Avg. | 70.7 | 66.6 |

training time of w/o Initial significantly increased. However, w/o Initial with these higher iteration numbers ($I = 10, 20, 30, 40, 50$) performed worse than the proposed method with $I = 3$. This would be because many iterations expand the computation graph of neural networks and it might makes the optimization difficult. This result shows the effectiveness of learning initial coefficient parameters.

### D.7 Experiments on Tasks with Different Response Spaces

In the main paper, we confirmed that the proposed method works well with the real-world datasets, in which the response (class) space is the same across tasks. However, the proposed method is also applicable to the case that the response space differs between tasks as described in Section B. We additionally evaluated the proposed method on Mnistr with different response spaces. In this experiment, for each task, we created a binary-classification problem by randomly choosing half of the 10 digits as positive and the rest as negative. That is, the positive (negative) classes for different tasks consist of different digits. Table 4 shows the results. In this experiment, we used the identity functions for task representation neural networks $u$ and $v$ in Section B and set the dimension of class-specific task representation $\mathbf{z}_c$ to two. Due to the difficulty of the problem, all methods degraded performance compared with the case of the same response space (Table 2 in the main paper). However, the proposed method outperformed the other methods. These results indicate that the proposed method works well even when the response spaces are different across tasks.

### D.8 Classification without Feature Selection

To evaluate the effectiveness of feature selection, we investigated classification performance without feature selection on the real-world datasets. Table 5 shows the results of SVM without feature selection (NoFS). The proposed method that trains SVM with selected features clearly outperformed NoFS on all datasets. This is because training data are too small to train SVMs with original high-dimensional features. Feature selection often improves performance when there are insufficient training data.

### D.9 Comparison with Multi-task Extensions of Lasso and HSIC Lasso

To further demonstrate the effectiveness of the proposed method, we considered multi-task extensions of Lasso and HSIC Lasso that use labeled data in both source and target tasks for training. We call these methods MLasso and MHLasso, respectively. Table 6 shows the results. Although MLasso and MHLasso were able to improve performance by using labeled data in source tasks, the proposed method performed better than MLasso and MHLasso on all datasets. Since MLasso and MHLasso are not designed for few-shot feature selection, they did not work well.

Table 6: Comparison with multi-task extensions of Lasso (MLasso) and HSIC Lasso (MHLasso): averages of test accuracy [%] over different numbers of target support instances and different numbers of selected features on each real-world dataset.

| Data | Ours | Lasso | HLasso | MLasso | MHLasso |
|---|---|---|---|---|---|
| Mnistr | 71.4 | 64.8 | 67.5 | 68.3 | 70.3 |
| Isolet | 68.0 | 61.9 | 60.3 | 67.0 | 65.9 |
| IoT | 72.6 | 71.4 | 68.7 | 69.1 | 69.2 |
| Avg. | 70.7 | 66.0 | 65.5 | 68.1 | 68.5 |

Table 7: Comparison on more difficult synthetic data: averages of fraction of correctly selected features [%] with different numbers of target support instances.

| $N_S$ | Ours | Lasso | HLasso |
|---|---|---|---|
| 10 | 37.7 | 19.3 | 24.7 |
| 20 | 49.0 | 30.7 | 43.3 |
| 30 | 59.4 | 32.7 | 51.7 |
| Avg. | 48.6 | 27.6 | 39.9 |

Table 8: Comparison on high-dimensional synthetic data ($D = 3000$): averages of fraction of correctly selected features [%] with different numbers of target support instances.

| $N_S$ | Ours | Lasso | HLasso |
|---|---|---|---|
| 10 | 39.7 | 12.6 | 8.0 |
| 20 | 46.7 | 18.3 | 22.6 |
| 30 | 53.7 | 22.7 | 27.7 |
| Avg. | 46.7 | 17.9 | 19.4 |

Table 9: Results with larger target support instances: averages of test accuracy [%] over different numbers of selected features and real-world datasets.

| $N_S$ | Ours | Lasso | HLasso |
|---|---|---|---|
| 50 | 73.4 | 70.5 | 70.0 |
| 80 | 75.1 | 72.9 | 75.2 |

### D.10 Simulation Experiments with More Difficult Settings

In the main paper, we used the synthetic data, in which the important feature indexes were selected from $\{(1, 2, 3), (4, 5, 6), \ldots, (28, 29, 30)\}$ in both source and target tasks. We additionally investigated the case that the important feature indexes of target tasks are randomly selected from $\{1, \ldots, 30\}$ such as $(3, 10, 26)$ and $(5, 7, 29)$. This is a more challenging but realistic setting. Table 7 shows the results. The proposed method clearly outperformed the other methods even in this difficult setting.

In addition, we investigated whether the proposed method works well on high-dimensional data. We created high-dimensional data by changing the feature dimension of the synthetic data described in the main paper from 100 to 3000. Table 8 shows the results. The proposed method clearly outperformed Lasso and HLasso by a large margin. Especially, although Lasso and HLasso significantly deteriorated the performance compared to the case with 100 features (Table 1 in the main paper) due to the curse of dimensionality, the proposed method was able to maintain the performance to some extent by meta-learning. These results indicate that the proposed method works well on high-dimensional data.

### D.11 Experiments with Larger Target Support Instances

This paper focuses on feature selection from a small number of target instances. However, it is interesting to investigate the performance of the proposed method when there are larger target instances. Table 9 shows the results with number of target support instances $N_S = 50$ and 80. When $N_S = 50$, the proposed method performed better the others. When $N_S = 80$, the proposed method and HLasso performed almost the same. This result is reasonable because when there is enough target data, the effect of source data generally becomes small.

# E   Limitations

The proposed method has a few limitations to overcome. First, the proposed method uses multiple source tasks to improve feature selection performance on unseen target tasks. However, when source and target tasks are significantly different, the performance on the target tasks risks degrading. This is known as "negative transfer," and overcoming it is one of the important problems in transfer/meta-learning studies. Developing methods to automatically remove negative effects of such tasks is a promising research direction.

Second, the proposed method assumes that the input feature space is the same across all tasks, which may hinder its practicality in some applications. However, this assumption is not unique to the proposed method but is common to almost all meta-learning methods [14, 13, 3, 2, 12, 7, 8]. Also, multi-task and transfer feature selection methods usually require the same assumption [10, 16, 1]. Even if the input feature space is the same, there are many possible practical applications. For example, as mentioned in Section 1, imagine that a company wants to analyze the features of its product ($\mathbf{x}$) that affect sales ($\mathbf{y}$) in each store/region (task). Within the same company, product $\mathbf{x}$ is often registered in the database using the same features (price, ingredients, etc). If we want to analyze important features from text documents in different tasks, the feature (word) space is identical within the same language. Developing meta-learning methods for feature selection that can handle different feature spaces is also a promising challenge.

Third, HSIC-based feature selection methods, including the proposed method, cannot capture some nonlinear feature interactions although HSIC can capture the nonlinear relationship between one variable $x$ and the response $y$. For example, if the predictive function is an XOR of two variables $x_1$ and $x_2$, HSIC cannot capture such interaction. Extending the proposed method to capture such relationships is also one of the important research directions.

# F   Social Impacts

The proposed method has some potential risks to be addressed when it is deployed to real-world applications. First, although we experimentally demonstrated the proposed method outperformed existing methods, it is not perfect; that is, it may select irrelevant features, which may lead to wrong analysis/interpretation in some cases. To mitigate this, people can use the proposed method as a support tool for their detailed analysis/interpretation of data. Second, the proposed method needs to access datasets obtained from multiple tasks like almost all transfer/meta-learning methods. When each dataset is provided from different owners such as companies, sensitive information in the dataset risks being stolen and abused by malicious people that use the proposed method. To evade this risk, we suggest promoting research for developing transfer/meta-learning without accessing raw datasets.