# OpenReview forum: "Few-shot Learning for Feature Selection with Hilbert-Schmidt Independence Criterion"
_NeurIPS.cc/2022/Conference — NeurIPS 2022 Accept_

### Official Review · Reviewer_WvPt · 2022-07-08

**Rating:** 6
**Confidence:** 2
**Soundness:** 3 good
**Presentation:** 4 excellent
**Contribution:** 3 good

**Summary:**

The paper proposes a meta-learning model for feature selection. The setup assumes the availability of labeled instances from several related tasks, and at test time, we only have access to a small number of labeled instances from the target task (called support set). A neural network infers kernel functions from the support set, and a Hilbert-Schmidt Information Criterion (a criterion for feature selection) based objective function is defined for optimization. The optimization problem is solved via the Iterative Shrinkage Thresholding (ISTA) algorithm. Results shown on several real-world datasets such as MNISTr, ISOLET, and IOT show the strong performance of the proposed algorithm.

**Questions:**

How are you able to learn efficient neural network representations for a task from the support set (Equation 4) since by definition you assume you only have access to a few samples only?

**Limitations:**

No potential negative social impacts of the research.

**Strengths And Weaknesses:**

++ The paper is well written and puts itself nicely in the context of the previous work. The proposed meta-learning approach appears novel, though it involves a complicated optimization procedure.

++ The experimental results are strong and show solid improvements over state-of-the-art baselines.

-- The paper assumes that only a few labeled observations are available from the support set. It is unclear how a highly parameterized neural network is reliably estimated from so few samples in Equation 4?

---

> ### Author Response · Authors · 2022-08-01
> **Reply to Reviewer WvPt**
>
> We want to thank the reviewer for the positive comments significantly!
>
> **> The paper assumes that only a few labeled observations are available from the support set. It is unclear how a highly parameterized neural network is reliably estimated from so few samples in Equation 4?**
>
> Thank you for your insightful question.
> The proposed method uses the meta-learning framework to learn how to select features from a few data using labeled data in multiple source tasks.
> The critical point is that the source tasks have sufficient labeled data to train the neural networks in Eq. (4).
> That is, since the neural networks are shared across all tasks and are meta-learned with all source labeled data, it can output good task representations $z$ even from a few labeled data without overfitting.
> We note that many few-shot learning (meta-learning) methods use such task representations (Eq. 4) and have shown effective performance [12,13,19,25].

---

> > ### Comment · Reviewer_WvPt · 2022-08-03
> > **Update after rebuttal**
> >
> >  I would like to thank the authors for a detailed rebuttal. My score stays the same, i.e., a weak accept. I only had a minor question in my review, which has been answered by the authors. It doesn't alter my overall evaluation of the paper.

---

### Official Review · Reviewer_GdFM · 2022-07-09

**Rating:** 5
**Confidence:** 4
**Soundness:** 2 fair
**Presentation:** 3 good
**Contribution:** 2 fair

**Summary:**

This paper applies the meta-learning to the HSIC feature selection, which can alleviate the problem of the lack of labeled data in some applications. It uses HSIC to measure the dependency of features and the labels, and then uses meta-training to train the proposed model. The experimental results show that the proposed method can outperform some feature selection methods, especially the non-meta-learning version HSIC Lasso.

**Questions:**

See above.

**Limitations:**

The authors discuss the limitation and social impacts in the Appendix. I think it's good.

**Strengths And Weaknesses:**

Strengths:
1. It applies meta-training to alleviate the problem of lack of labeled data, and the experimental results show the effectiveness of meta-learning.
2. The experimental results are good.

Weaknesses:
1. The paper claims that this is the first work of meta-learning for supervised feature selection. However, the idea that applying meta-learning for feature selection has been already proposed before, e.g. [1]. Although the methods of these two papers are quite different, the idea of combining meta-learning and feature selection is not new.
2. The main part of the proposed method except the meta-learning is very similar to HSIC Lasso [2], e.g. the objective function and the interpretation of HSIC are similar. It seems that this paper is an extension of [2] by adding the standard meta-learning. The paper does not show any innovative thing about meta-learning and just applies it to the feature selection.
3. In the experiments, the used data sets are not enough. There are only three real-world data sets. However, in the literature on feature selection, often six to ten data sets are used.
4. Feature selection is often used to handle high-dimensional data. However, none of the data sets used in this paper are high dimensional.
5. It would be better to show the classification results on all features, which can show whether feature selection is useful for the classification.
6. The Conclusion part is too short. It would be better to summarize the strengths and limitations of the proposed method and introduce some future work.

[1] Meta-Learning Causal Feature Selection for Stable Prediction, in ICME 2021
[2] High-dimensional feature selection by feature-wise kernelized lasso, in Neural Computation 2014.

---

> ### Author Response · Authors · 2022-08-01
> **Reply to Reviewer GdFM (1/2)**
>
> We greatly thank the reviewer for the constructive and insightful comments!
>
> **> 1. The paper claims that this is the first work of meta-learning for supervised feature selection. However, the idea that applying meta-learning for feature selection has been already proposed before, e.g. [1]. Although the methods of these two papers are quite different, the idea of combining meta-learning and feature selection is not new.**
>
> Thank you for sharing this exciting and relevant paper.
> As you mentioned, the method in [1] also uses a meta-learning framework,
> but there are apparent differences between [1] and our paper.
>
> First, the purposes of the papers are different. The proposed method aims to select a "task-specific" feature subset from "the original features" with a few labeled data.
> By selecting task-specific features from the original ones,
> we can interpret/analyze each dataset (task) and eliminate the cost of collecting irrelevant/redundant features.
> On the other hand, the method in [1] aims to select a "task-invariant" (stable) subset from "hidden features" obtained through CNNs for stable prediction.
> Since this method treats hidden features, it looks more like a feature extraction method than feature selection (thus, this method does not provide benefits such as interpretability and cost reduction as mentioned above). In addition, it does not seem to be aimed at learning from a small number of labeled data (e.g., N = 10).
>
> Second, as you commented, the methods are entirely different.
> The method in [1] uses only NNs to learn the task-invariant mask (just learnable parameters) to obtain the stable hidden feature subset.
> On the other hand, the proposed method combines the kernel-based approach (HSIC Lasso) and neural networks.
> Tables 1 and 2 demonstrate the effectiveness of this approach.
>
> In the final paper, we would like to include the reference [1] and the above discussion in the related work section.
> In addition, to clarify that the proposed method is designed for few-shot feature selection, we will consider changing the description of the proposed method to a method for "few-shot learning" instead of "meta-learning."
> In particular, we will rewrite the sentence "our work is the first attempt at meta-learning for supervised feature selection with a small number of labeled instances." in Line 38--40 into "our work is the first attempt at few-shot learning for supervised feature selection".
>
> **> 2. The main part of the proposed method except the meta-learning is very similar to HSIC Lasso [2], e.g. the objective function and the interpretation of HSIC are similar. It seems that this paper is an extension of [2] by adding the standard meta-learning. The paper does not show any innovative thing about meta-learning and just applies it to the feature selection.**
>
> As acknowledged by the other three reviewers, the significant contribution of this paper is the integration of techniques from different areas, HSIC Lasso and meta-learning, to address the critical and challenging problem of "feature selection from small amounts of data" and to demonstrate their effectiveness experimentally.
>
> In addition, as mentioned in Section 2, our meta-learning design has several advantages over simply applying well-known meta-learning methods such as MAML [9] and neural processes [12,13,19] to feature selection. Specifically, it would be more efficient than MAML by performing support set adaptation with the simple closed-form update steps of only feature importance vector $\alpha$ Eqs. (6,7) instead of adapting the whole parameters of the neural networks with gradient descents as in MAML. In addition, the adaptation directly minimizes the loss of the support set, so it can be adapted more effectively than neural processes that do not explicitly minimize the loss, as demonstrated in our experiments.
>
> **> 3. In the experiments, the used data sets are not enough. There are only three real-world data sets. However, in the literature on feature selection, often six to ten data sets are used.**
>
> Thank you for your good comments.
> Many feature selection studies for treating multiple tasks use about one to three real-world datasets [37,56,2,a,b].
> This is because each real-world dataset encompasses multiple tasks.
> Thus, in this paper, we also used three real-world datasets that have multiple tasks.
> Also, in each dataset, many experiments were conducted under various conditions, varying the number of selected features and support set sizes (Table 2).
> Nevertheless, as you commented, we want to evaluate the proposed method with other real-world datasets.
>
> [a] Zhang, Yu, Dit-Yan Yeung, and Qian Xu. "Probabilistic multi-task feature selection." Advances in neural information processing systems 23 (2010).
>
> [b] Hernández-Lobato, Daniel, José Miguel Hernández-Lobato, and Zoubin Ghahramani. "A probabilistic model for dirty multi-task feature selection." International Conference on Machine Learning. PMLR, 2015.

---

> > ### Author Response · Authors · 2022-08-01
> > **Reply to Reviewer GdFM (2/2)**
> >
> > **> 4. Feature selection is often used to handle high-dimensional data. However, none of the data sets used in this paper are high dimensional.**
> >
> > We have evaluated the proposed method in case the feature dimension is high ($D=3000$) in Appendix E.11.
> > The proposed method could select features much more accurately than other methods (Table 11).
> > This result suggests that the proposed method works well on high-dimensional data.
> >
> > **> 5. It would be better to show the classification results on all features, which can show whether feature selection is useful for the classification.**
> >
> > Thank you for your constructive feedback.
> > To investigate the effectiveness of feature selection,
> > we have performed an experimental comparison with the method with all features (NoFS) in Appendix E.9.
> > NoFS performed worse than the proposed method that trains SVM with selected features (Table 8).
> > This is because training data are too small to train SVMs with original high-dimensional features
> > (the so-called "curse of dimensionality" problem).
> > This result demonstrates the effectiveness of feature selection using the proposed method.
> >
> > **> The Conclusion part is too short. It would be better to summarize the strengths and limitations of the proposed method and introduce some future work.**
> >
> > Thank you for your suggestion. In the conclusion of the final version, we want to summarize the strengths of the proposed method and the limitation described in Appendix F in the current manuscript. In future work, we plan to combine HSIC Lasso and neural network-based classifiers in our meta-learning framework to simultaneously perform feature selection and classifier learning from a few labeled data.

---

> > > ### Comment · Reviewer_GdFM · 2022-08-10
> > > **Response to authors**
> > >
> > > Thanks for the rebuttal. After reading the rebuttal, I would like to rise my score.

---

### Official Review · Reviewer_mXfb · 2022-07-10

**Rating:** 6
**Confidence:** 4
**Soundness:** 3 good
**Presentation:** 3 good
**Contribution:** 3 good

**Summary:**

The authors address a challenging and important problem of feature selection. They focus on selecting the most informative features given samples from multiple source tasks, and a few target labeled observations. This is particularly challenging if the number of features is large and if there is a distribution shift between the source and the target data. They use Hilbert-Schmidt Independence Criterion to measure dependencies between features and the label. The proposed solution relies on meta-learning with iterative optimization.  Finally, the authors use synthetic and real data to compare the proposed approach to existing schemes.

**Questions:**

The first few sentences of the second part of the introduction are not written very well. Specifically, the motivation for supervised fs and its relation to unsupervised fs could be explained by more details and better motivation of the problem
Figure 2 requires clarification, specifically, what features are selected exactly by \alpha, it seems that more than the first two (so the FDR is not low).
Can you provide a comparison to STG when it is trained end to end with the predictive model, i.e. without kernel SVM?
I might be missing something, but I don’t see where in algorithm 1 you use the small set of labeled samples given from the target dataset, why?


**Limitations:**

The limitations of the proposed work are properly detailed in section F.

**Strengths And Weaknesses:**

The paper reads well, and the English level is satisfactory. The problem of feature selection with few samples and multiple domains seems unique, and I haven’t seen many works address this regime before. The use of meta-learning feels adequate for this problem, and I like the framework proposed by the authors. Another strength of the method is the use of permutation invariant networks, which probably help the method succeed in a regime of low sample size. One key limitation is the use of HSIC, which is a strong statistical tool, but as I understand doesn’t take into account nonlinear feature interactions. For example, if the predictive function is an XOR of x1 and x2, how would FS mechanism that uses HSIC with univariate feature nonlinearity capture such interaction? I think this is a major drawback, that should at least be mentioned and explained to the reader.
Another concern is the evaluation criteria, the authors compare all methods using a kernel-based classifier. Since STG is an NN-based embedded method it should be evaluated as such, the features selected by the method are most appropriate for a NN classifier (specifically for the one learned after convergence), so it seems weird to evaluate the accuracy of the model using kernel SVM.

---

> ### Author Response · Authors · 2022-08-01
> **Reply to Reviewer mXfb (1/2)**
>
> We want to thank the reviewer for the positive and insightful comments significantly!
>
> **> One key limitation is the use of HSIC, which is a strong statistical tool, but as I understand doesn't take into account nonlinear feature interactions. For example, if the predictive function is an XOR of x1 and x2, how would FS mechanism that uses HSIC with univariate feature nonlinearity capture such interaction? I think this is a major drawback, that should at least be mentioned and explained to the reader.**
>
> This is an insightful comment.
> While the effectiveness of HSIC-based feature selection methods has been reported in many studies [53,18,11,24,47],
> this approach seems to have difficulty capturing nonlinear feature interactions, such as the XOR example in your comment. However, this is true not only for HSIC-based methods but also for other well-known feature selection methods such as Lasso.
> We think extending the proposed method to work in such cases would be good future work.
> We will discuss this point in the Limitation section of the final version.
>
> **> Another concern is the evaluation criteria, the authors compare all methods using a kernel-based classifier. Since STG is an NN-based embedded method it should be evaluated as such, the features selected by the method are most appropriate for a NN classifier (specifically for the one learned after convergence), so it seems weird to evaluate the accuracy of the model using kernel SVM. Can you provide a comparison to STG when it is trained end to end with the predictive model, i.e. without kernel SVM?**
>
> Thank you for your suggestion.
> HSIC-based feature selection methods are classifier-independent feature selection methods,
> i.e., they can perform feature selection without using any classifiers.
> Since we do not know true important features in real-world datasets,
> to assess the selected features with these methods,
> additional classifiers such as SVM are often used for evaluation as described in Lines 294--299 [43,47,29,24].
> Thus, we used the SVM for all methods to assess the quality of the selected features fairly.
> We did not use neural networks as classifiers since they have many hyperparameters to be tuned, and target training data are too small to train without overfitting.
> We note that as discussed in Introduction and Appendix A, the lack of a classifier would not be a significant drawback of the proposed method since the feature selection function, not the classification, is often essential in real applications.
>
> In addition, we note that the proposed method outperformed STG, STG-S, and STG-FT in the synthetic regression data, where any regressors (predictors) were not used since important features were known in the dataset (thus, we evaluate the quality of the selected features directly).
> This result (Table 1) indicates that the proposed method has better feature selection ability than other methods without using any predictors.
>
> We think that combining HSIC Lasso and neural network-based classifiers in a meta-learning framework is one of the promising research directions for few-shot feature selection.
>
> **> The first few sentences of the second part of the introduction are not written very well. Specifically, the motivation for supervised fs and its relation to unsupervised fs could be explained by more details and better motivation of the problem**
>
> Thank you for your constructive suggestion.
> In many practical applications, we want to select features responsible for predicting a response (label).
> This is the case, for example, when we want to find a gene (feature) associated with the presence of a disease (label) or a product attribute (feature) associated with sales (label).
> Since unsupervised feature selection methods do not use label information,
> it is difficult to select features associated with such labels.
> On the other hand, supervised feature selection methods can directly identify features related to labels by using labeled training data.
> This is why we focus on supervised feature selection in this study.
> In the final paper, we would like to describe more clearly the motivation for supervised feature selection and the relationship with unsupervised one as described above.
>
> **> Figure 2 requires clarification, specifically, what features are selected exactly by \alpha, it seems that more than the first two (so the FDR is not low)**
>
> The proposed method selects the top-K features by sorting the values of learned $\alpha$ in descending order as described in Lines 137--139. This procedure is equivalent to the original HSIC Lasso [53].
> Figure 2 shows that the top-3 values of $\alpha$ are for feature indices 1, 2, and 3.
> Since feature indices, 1, 2, and 3 are important features in the synthetic data used in this experiment,
> this result indicates that the proposed method could correctly select important features.
> We will add the above explanation to the caption of Figure 2 in the final paper.

---

> > ### Author Response · Authors · 2022-08-01
> > **Reply to Reviewer mXfb (2/2)**
> >
> > **> I don't see where in algorithm 1 you use the small set of labeled samples given from the target dataset, why?**
> >
> > We are sorry for the lack of explanation.
> > Algorithm 1 is a pseudo-code for the "training" procedure using data from the "source tasks."
> > After training the model with Algorithm 1, we can obtain target task-specific feature importances $\alpha$ from a few target labeled data $S$ using the learned model (specifically, by running lines 5 through 9 of Algorithm 1 with the target data $S$).
> > We want to clarify this point in the final paper.

---

> > > ### Comment · Reviewer_mXfb · 2022-08-08
> > > **Response to authors**
> > >
> > > I thank the authors for addressing my comments. The clarifications are helpful and should be incorporated into the paper. I agree that the HSIC is a valuable statistical tool and very useful for feature selection. Nonetheless, extending your method to allow for nonlinear feature interactions seems like a promising future direction. I think that the paper is important and contributes to the community and keep my score at acceptance, good luck!

---

### Official Review · Reviewer_wGbm · 2022-07-10

**Rating:** 7
**Confidence:** 4
**Soundness:** 3 good
**Presentation:** 2 fair
**Contribution:** 3 good

**Summary:**

The authors tackle the problem of efficient feature selection on small datasets when having access to related datasets. They propose to encode information about each dataset via a small random sample of instances (e.g. 10-30). The approach then defines a set of parametrised gaussian kernels (one per dimension) where the parameters (the importance of each feature and the scaling for each feature) are learned using a permutation invariant neural network allowing to model non linear dependencies between the features and the target.

**Questions:**

The paper would gain strength if the authors could show
1) a comparison with the baseline proposed in Neg 2), i.e. compare against the baseline of training a permutation invariant (deep) neural network on the task of learning the feature selection mask directly.
2) a statistical test for the ablation study, perhaps the one suggested in neg 3).

**Limitations:**

yes.

**Strengths And Weaknesses:**

Pos
Given the known HSIC Lasso approach to define the optimization objectives, the authors propose a ANN architecture to learn the few (2) parameters of feature-wise kernels which allows them to be sample efficient.

Neg
1) The introduction and layout of the material could probably be improved to streamline the introduction of the necessary concepts in an orderly fashion.

2) The methods relies heavily on the capacity of the ANN to generalise le selection of features across related problems. To better clarify the contribution of the approach one should also compare against the baseline of training a permutation invariant (deep) neural network on the task of learning the feature selection mask directly, i.e., running some feature selection approach to obtain which features to select for each dataset. Note that in this case the feature selection performed for each task could fully exploit all the instances in the respective datasets and obtain better feature importance estimates.

3) The ablation study is important and well conceived. It would however be of interest to be able to show which element is yielding significant differences (as each ablation seems to be within the 1% range). Perhaps performing some statistical test using the approach explained in [Janez Demšar. 2006. Statistical Comparisons of Classifiers over Multiple Data Sets. J. Mach. Learn. Res. 7 (December 2006), 1–30] and readily available as a python package in [https://github.com/sherbold/autorank].

---

> ### Author Response · Authors · 2022-08-01
> **Reply to Reviewer wGbm**
>
> We greatly thank the reviewer for the positive comments and constructive feedback!
>
> **> 1. The introduction and layout of the material could probably be improved to streamline the introduction of the necessary concepts in an orderly fashion.**
>
> Thank you for your constructive advice.
> We will revise the introduction and layout of the material in the final version to improve clarity.
>
> **> Q1. a comparison with the baseline proposed in Neg 2), i.e. compare against the baseline of training a permutation invariant (deep) neural network on the task of learning the feature selection mask directly.**
>
> Thank you for your suggestion.
> The comparison method (SMetaGS) in the submitted paper can be regarded as the method you suggested.
> SMetaGS is a neural network-based meta-learning method for supervised feature selection that meta-learns feature selection mask $M \in \mathbb{R}^{D}$.
> Specifically, this mask is modeled by a permutation-invariant NN that takes support set (a few labeled data) $S$ as input (i.e., $M(S)$) and is meta-learned to improve test feature selection performance using the source labeled data.
> After meta-learning, feature selection on a target task is performed by inputting a few target labeled data into the trained permutation-invariant NN.
> The proposed method performs much better feature selection than SMetaGS (Tables 1 and 2).
> This result demonstrates the difficulty of meta-learning with only NNs, which is described in Lines 100--109,
> and the effectiveness of the HSIC-based formulation of the proposed method.
>
> In addition, other neural network-based comparison methods (STG, STG-S, STG-FT) also learn masks for feature selection.
> Since these methods treat mask $M$ as just training parameters, they cannot change the values of the mask depending on the support set (task).
> STG, STG-S, and STG-FT are methods that use only target data, only source data, and both target and source data, respectively.
> The proposed method outperforms these methods in feature selection performance (Tables 1 and 2).
>
> We want to add a detailed description of these comparison methods in the final paper to improve the clarity.
>
> **> Q2. a statistical test for the ablation study, perhaps the one suggested in neg 3).**
>
> Thank you for your insightful suggestion.
> To investigate the significant differences,
> we performed the paired t-test with the significance level of $5\\%$ for each dataset.
> The number of best or comparable results are as follows:
>
> | Ours | w/o Latent | w/o Feature | w/o LFeature | w/o S-kernel | w/o Initial | w/ Deep | w/o S-adapt |
> | ---: | ---: | ---: | ---: | ---: | ---: | ---: | --: |
> | 3 | 1 | 1 | 0 | 1 | 1 | 1 | 1 |
>
> We confirmed that the proposed method statistically performs better than other variants.

---

### Meta-Review · Area_Chair_RoPu · 2022-08-27

**Recommendation:** Accept
**Confidence:** Less certain

**Metareview:**

The  paper studies the problem of feature selection with few labelled samples. The paper develops an optimization framework which applies for both regression and classification: the regression setting has been explored for exposition and the classification details are in the appendix. The  use of permutation invariant Neural networks  and using Multi-task learning are interesting angles which help
the paper demonstrate that features can be selected even when the number of labelled examples are small.




**Award:**

No

---

### Decision · Program_Chairs · 2022-09-14

Accept